# Friction and Wear Performance of CoCrFeMnNiW Medium-Entropy Alloy Coatings by Plasma-Arc Surfacing Welding on Q235 Steel

**Qingxian Hu, Xiaoli Wang \*, Junyan Miao, Fanglian Fu and Xinwang Shen**

School of Materials Science and Engineering, Jiangsu University of Science and Technology, Zhenjiang 212003, China; huqingxian@just.edu.cn (Q.H.); 182060002@stu.just.edu.cn (J.M.); flfu1997@stu.just.edu.cn (F.F.); 1440602222@stu.just.edu.cn (X.S.)

\* Correspondence: xlwang@just.edu.cn

**Abstract:** In this study, CoCrFeMnNiW medium-entropy alloy coating on Q235 was fabricated by plasma surfacing technology. The wear performance of the prepared one-layer coating and the two-layer coating was studied by a friction and abrasion tester. The microstructure and performance of the CoCrFeMnNiW coating were researched by optical microscope, a nano-indentation test, SEM, and hardness tester. The results show that the microstructure of the coating is made up of a fusion zone, equiaxed dendrites near the fusion zone, coarse columnar crystals, and near-surface with a certain direction between the near-fusion zone and near-surface fine equiaxed grains. The wear mechanism of one layer coating was abrasive with wear and fatigue wear. The wear mechanism of the two-layer coating was adhesive with wear and fatigue wear. For CoCrFeMnNiW MEA coating, the main factors determining their wear resistance were the value of its depth recovery ratio ($\eta_h$) and EIT.

**Keywords:** welding layer; plasma; microstructure; friction and abrasion; medium-entropy alloys

## 1. Introduction

Plasma-arc surfacing welding is widely used in part surface abrasion-resistant and corrosion-resistant alloy welding layers because of its many advantages, such as high energy concentration and penetration, small deformation [1], low dilution rate [2], good controllability [3], high thermal efficiency [4], good formability [5], and so on. With the development of industry, the requirements for wear resistance of materials are higher and higher. The materials with superior wear resistance have been in demand for the past few decades.

High/medium-entropy alloy (HEA/MEA) materials have become extensively investigated by many researchers in recent years [6,7]. HEA/MEA are composed of multiple main elements. This new alloy greatly expanded the scope of alloy systems, and it increases the possibility of improving its performance, such as high strength and excellent wear performance. However, because it contains noble alloying elements, the cost of producing bulk HEA/MEA by casting may be expensive. Until now, many studies have been focused on the preparation and properties of HEA/MEA coating. The preparation methods of HEA/MEA coating include the resistance welding method [8], utilizing laser cladding [9–11], DC magnetron sputtering [12,13], and plasma spray [14], etc. Among them, laser cladding is widely used to fabricate HEA/MEA coatings [15,16]. Owing to its advantages of high temperature and the fast cooling rate, it is considered an ideal method to prepare HEA/MEA coatings. In this process, the laser is its heat source and power is its filling materials; laser equipment is more expensive. Because the laser spot beam is small, its efficiency is relatively low. In its practical engineering application, material performance, as well as material cost and material processing, should be considered [17,18]. Compared

with laser cladding, plasma-arc surfacing welding has obvious economic characteristics of low cost and high efficiency. However, few studies have been carried out on the friction and wear performance of HEA/MEA coating by plasma-arc surfacing welding. Therefore, it is interesting to fabricate a HEA/MEA coating by plasma arc welding.

CoCrFeNiW MEA recently became a research hotspot because it is considered as the binder phase to replace Co in cemented carbides (WC-Co). Cai et al. fabricated FeCoCrNiCux coatings on Cr12MoV die steel by laser cladding technology [19]. The experimental materials were HEA power with a purity of 99.5–99.5% and granularity of about 50–80 μm. It was found that the FeCoCrNiCux cladding layers belong to MEA coating because of the dilution effect of the substrate [20]. The problem of the dilution effect was unavoidable in the process of preparing coating by surface cladding. In this study, the MEA coatings were manufactured by using the wire, which was relatively cheaper than the high entropy alloy powder by making full use of the effect of the dilution rate.

In the process of plasma surfacing welding, due to the dilution effect of the substrate, Fe element entered into the coating from the substrate, which resulted in the change of the proportion of each element in the coating. MEA coating can be prepared by selecting suitable welding wire.

To compare the effect of this change on the microstructure and properties of the coatings, one-layer and two-layer coatings were prepared. The microstructure, hardness, and wear resistance of one-layer and two-layer coatings prepared by plasma-arc surfacing welding technology were analyzed in this study.

## 2. Materials and Methods

GH605 welding wire and the substrate Q235 were used in this work. GH605 was a kind of cobalt-based wire. Q235 was a common carbon structural steel with good toughness and plasticity, good weldability and thermal machinability. Table 1 shows the main chemical compositions of GH605 welding wire and Q235. The plasma-arc welding machine was Trans Tig5000 Series Digital automatic welding machine made by Austria Fronium. Figure 1 was the process diagram of surfacing welding coating preparation. Each number, 1, 2, 3, 4, 5, represents a welding pass. When the first welding pass 1 was finished, it was cooled in the air for 5 min and the second welding pass 2 was overlaid in the same direction. Between the first welding pass and the second welding pass, there was an overlap area, and it covered one-third of the welding pass. The third welding pass repeated the process. Several welding passes were conducted, and then a one-layer coating was obtained. When the one-layer coating was finished and cooled to room temperature, the second layer was added to the surface of the first layer in the same method. In this way, the one-layer coating and two-layer coating were abstained. The surfacing welding process parameters were shown in Table 2. The protective gas used throughout the process was argon with a purity of 99.99%.

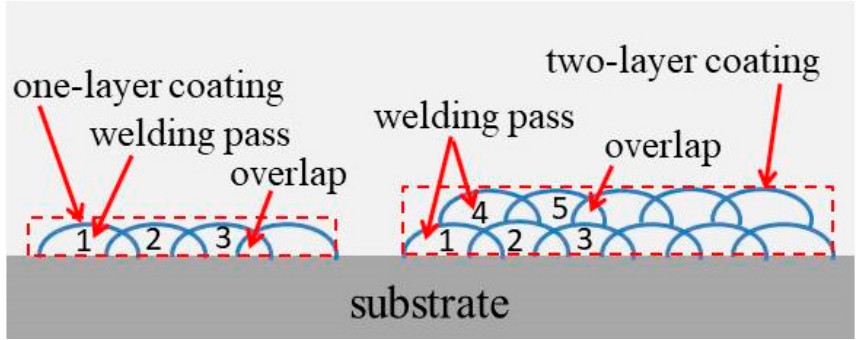

**Figure 1.** Process diagram of surfacing welding coating preparation.

**Table 1.** The main chemical composition of the experimental materials (wt.%).

| Element | Mn | Si | C | Fe | Cr | Ni | W | Co | P | S |
|---|---|---|---|---|---|---|---|---|---|---|
| GH605 | 1.0~2.0 | ≤0.40 | 0.05~0.15 | 3.00 | 19.0 | 9.0~11.0 | 14.0~16.0 | Bal. | 0.04 | 0.03 |
| Q235 | 0.30 | 0.15 | 0.17 | Bal. | - | - | - | - | 0.015 | 0.035 |

**Table 2.** The parameters of the plasma welding process.

| Parameters | Welding Voltage (V) | Welding Current (A) | Welding Speed (cm/min) | Welding Torch Height (mm) | The Plasma Gas Flow (L/min) |
|---|---|---|---|---|---|
| Value | 20.1 | 125 | 24 | 7 | 2.5 |

The CoCrFeMnNiW coatings fabricated on the surface of Q235 was shown in Figure 2. The coating of A and B were one-layer and two-layer surfacing layers, respectively. The parts of A1 and B1 were tested for friction and wear performance. The parts of A2 and B2 were observed microstructure. The specimens were grinded by 60 meshes, 240 meshes, 600 meshes, 800 meshes, 1000 meshes sandpapers in turns, mechanically polished with 1.5 μm diamond paste, and then electrolytic etched in 10% CrO$_3$ water solution for 50–60 s. The parts of A3 and B3 were used for the hardness test.

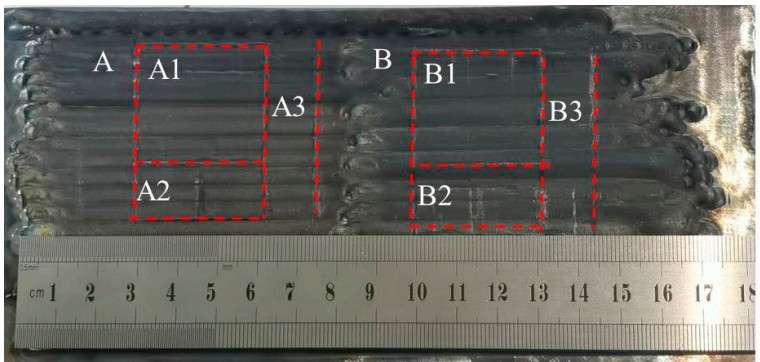

**Figure 2.** Specimen areas dedicated for wear (A1, B1), microstructure (A2, B2) and hardness (A3, B3) testing. coating A: one-layer, coating B: two-layer coating.

The chemical composition of the two coatings was examined by a PDA5500II direct-reading spectrograph (SHIMADZU, Kyoto, Japan). The metallographic microstructure of Co-based welding layers was observed using a Nikon Epiphot 300 optical microscope (Nikon, Tokyo, Japan). A SU-70 field emission scanning electron microscope (Hitachi, Tokyo, Japan) with an energy dispersive spectroscope (EDS) was applied to study the distribution of elements and wear morphology.

The elastic performances of materials are the basements to determine the behavior of materials under stresses and loads. In order to test the elastic property of the surfacing layers, nanoindentation curves were obtained by a nanoindenter of the CSM NHT2 (Anton Paar) model with a Berkovich tip made in Switzerland. The maximum load was 5 mN, and the loading/unloading rate was 10 mN·min$^{-1}$. The elastic property of welding layers can be characterized by its depth recovery ratio ($\eta_h$) drew from the load-displacement curve. Generally, the material with a higher $\eta_h$ value had a better elastic property. The calculation formula of $\eta_h$ is as follows [21]:

$$\eta_h = \frac{h_{max} - h_r}{h_{max}} \tag{1}$$

where $h_{max}$ was the maximum indentation depth (nm) and $hr$ was the residual depth after unloading (nm). The elastic modulus (EIT) of the coatings can be obtained directly by the test. These experiments were repeated five times. The test result was the average value of

five experiments. The ranges between plus and minus were obtained by calculating the standard deviation of five experimental data.

The DHV-1000 model of hardness tester (Shanghai Shangcai Tester Machine Co., LTD., Shanghai, China) was used to test Vickers hardness of the coatings A and B. The hardness was marked every 0.25 mm. The distance was used as the horizontal axis, and the hardness value was used as the vertical axis. The test load was 10 N and the last time was 10 s.

The HSR-2M friction tester (Lanzhou Institute of Chemical Physics, Chinese Academy of Sciences, Lanzhou, China) was used to evaluate the wear performance of the two coatings. The ceramic ball ($Si_3N_4$) was taken as the friction couple, and the coating samples were the working disk. The vertical load was applied to the working disk and the transverse load made the $Si_3N_4$ ball to reciprocate at 5 mm in a straight line at a sliding speed of 25 mm/s for 30 min, without lubrication and in the air. The loads of the tests were chosen 10 N, 15 N and 20 N, respectively. In the friction process, the data was transmitted to the computer by the sensor on the loading rod, and the corresponding friction coefficient was calculated. After the friction test, the morphologies of wear scars and cross-section were observed using a laser scanning confocal microscope (LEXT, OLS400 LSCM, Tokyo, Japan) and SEM. The value of $W_s$, namely the wear rates ($mm^3 \cdot N^{-1} \cdot m^{-1}$), can be calculated by the following formula [21]:

$$W_s = \frac{CA}{FL} \tag{2}$$

In which, $C$ is the width of the wear scar (mm), $A$ is the average wear area of wear loss ($mm^2$), $F$ is the applied load (N), and $L$ is the distance of sliding friction (mm).

## 3. Results

### 3.1. Material Characterization

Figure 3 shows optical micrographs of the coatings A and B, respectively. As was seen from Figure 3a and the enlarged graphs, there were four different parts of the zone from bottom to top. They were the fusion zone, the equiaxed dendrite near the fusion zone and coarse dendrite with a certain direction between the near-fusion zone and the near-surface zone, and fine equiaxed crystals next to the surface.

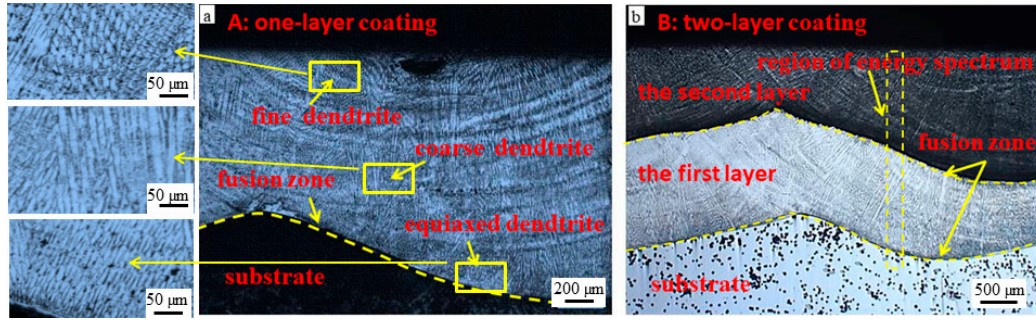

**Figure 3.** Optical micrographs of coatings A and B and the region and direction of energy spectrum analysis. (**a**) is the optical micrograph of coating A. (**b**) is the optical micrograph of coating B.

According to the theory of crystallization, in extremely cold conditions, the crystallization process of the surfacing welding molten pool was a kind of heterogeneous nucleation [22]. Under the effect of the plasma arc of a high energy heat source, the liquid composition fluctuated, the cooling rate of the local area was different, and the organization of different zones presented obvious inhomogeneity. With the passage of the solidification interface, the crystal nucleus rapidly grew because the liquid temperature gradient decreased and the super-cooling composition increased. Therefore, the equiaxed dendrite near the fusion zone formed, and the coarse dendrite that had a certain direction between the facing layer and near the fusion-zone area was generated. It can also be found that with the increase of the distance from the fusion zone, the grain becomes finer and finer, and the

columnar dendrite transforms to an equiaxed crystal. Figure 3b was the metallography of the two-layer coating. It was seen that organization was given priority to the coarse dendrite. From the bottom to the top, dendrite became thicker, and the distance between dendrites increased.

Table 3 shows the chemical composition of coatings A and B. It was shown they were the CoCrFeMnNiW alloy coatings. The difference in chemical composition between A and B was reflected in the content of Fe. The Fe content in coating A increased by 47.43% compared with that of coating B. Obviously, this part of Fe came from the substrate. The content of other elements, such as Co, Cr, W in coating A was lower than that in coating B.

**Table 3.** Analysis results of the chemical composition on the surface of coatings A and B (mole fraction, %).

| Coating | Co | Cr | Fe | W | Ni | Mn |
|---------|-------|-------|-------|------|------|------|
| A | 39.11 | 17.44 | 30.06 | 3.59 | 8.20 | 1.60 |
| B | 47.36 | 20.93 | 15.80 | 4.41 | 9.82 | 1.68 |

According to the Boltzmann hypothesis, the entropy value ($\Delta S$) is [23]:

$$\Delta S = -R[X_1 ln X_1 + X_2 ln X_2 + \ldots X_n ln X_n] = -R\sum_{i=1}^{n} X_i ln X_i \quad (3)$$

where $R$ is a gas constant, $X_i$ is the molar ratio of principal element, n is the concentration of mixing elements. Alloys are divided into high-, medium- and low-entropy alloys according to the value of $\Delta S$. High entropy is defined as have $\Delta S \geq 1.5\,R$. For medium-entropy alloy, the value of $\Delta S$ is from $R$ to $1.5\,R$. For the low-entropy alloy, $\Delta S \leq R$ [23].

According to Formula (3), the mixing entropy of the alloying layers system can be calculated by the molar fraction of each element. The mixed entropy of coating A and B was $1.42\,R$ and $1.40\,R$, respectively. The surfaces of coatings A and B were the CoCrFeMnNiW medium-entropy coating.

To study the distribution of elements in the cross-section, the energy spectrum analysis of coating B from the coating surface to the substrate (direction) was carried out. The results were shown in Figure 4. On the one hand, because of the dilution effect from the substrate, it can be seen that the content of the Fe element in the medium layer was obviously enhanced. Because the content of Fe is high in the substrate and the top layer was far away from the substrate, the change of Fe content shows a decreasing trend from the substrate to the top of the coating. On the other hand, due to the high content of Co and Cr in the welding wire GH605, the content of Co and Cr increases step by step from the substrate to the top of coating B. Because of the relatively low content of W, Ni and, Mn in the welding wire GH605, their increasing step trend is not obvious from the substrate to the top of coating B. The entropy values of all points were calculated. The entropy values in cross-section of the intermediate coating and the top coating were between $1.42\,R$ and $1.47\,R$, and between $1.39\,R$ to $1.49\,R$, separately. Therefore, combined with the analysis of entropy calculation results of the coating surface, it can be determined that both A and B coatings belonged to the medium-entropy alloy coating.

It can also be seen from Figure 4 that the contents of Co and Cr in the alloy coating were high. The solid solution can be formed between the element Co and Cr. The elements of Ni and W will also solubilize in the solid solution, playing a role in solid solution strengthening. In particular, the solid solubility of the W element in the cobalt-based solid solution matrix is large, and its atomic radius is larger than those of other elements. Therefore, the solution of W in the matrix causes an obvious lattice deformation, resulting in solid solution strengthening. This is a typical character of HEA and MEA with multi-components and has been reported previously, which is called the effect of lattice deformation [24]. The contents of Co, Cr, Ni and W in the top coating are higher than that in the medium coating so that

the solid solution strengthening effect of alloy elements is strengthened, which may lead to a higher hardness of the top coating [25]. In other words, the hardness of the two-layer coating B is higher than that of the one-layer coating A.

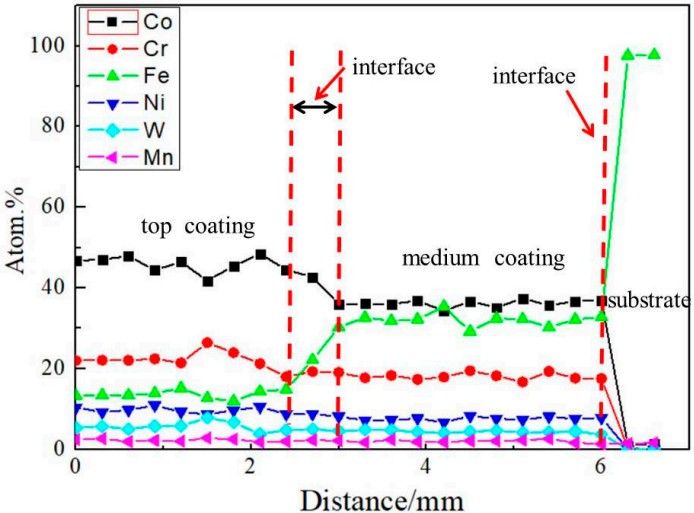

**Figure 4.** The distribution of elements in cross-section of coating B.

### 3.2. Nanoindentation Test

Figure 5 shows the load-displacement curve of the coatings A and B in the nanoindentation test. The maximum penetration depth ($h_{max}$) of coatings A and B at a load of 5 mN was 182.47 nm and 161.87 nm, and the residual depth ($h_r$) of A and B specimen after unloading was 161.87 nm and 145.55 nm, respectively. According to Equation (1), the values of $\eta_h$ were calculated to be 0.11, and 0.10, respectively.

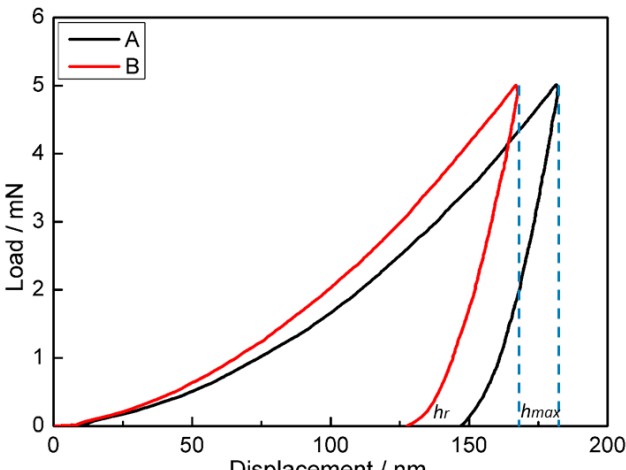

**Figure 5.** Load-displacement curves of the alloy coating A and B.

The value of $\eta_h$ for coating A was higher than that for coating B. It indicated that A coating had a higher resistance to plastic deformation than B coating. It also indicated that the value of $\eta_h$ and EIT was higher. Then, under the load, the contact area between the friction couple and the coating was larger, resulting in the higher friction coefficients.

The indentation parameters derived from the load-displacement curve in Figure 4 was shown in Table 4.

**Table 4.** Indentation parameters derived from the load-displacement curve in Figure 3.

| Specimen | $h_{max}$ (nm) | $h_r$ (nm) | $\eta_h$ | EIT (GPa) |
|----------|----------------|-------------|-----------------|-------------------|
| A | $182.47 \pm 1.38$ | $161.87 \pm 1.13$ | $0.11 \pm 0.003$ | $271.01 \pm 1.76$ |
| B | $161.87 \pm 1.26$ | $145.55 \pm 1.06$ | $0.10 \pm 0.002$ | $266.29 \pm 1.51$ |

### 3.3. Wear Performance

Figure 6 shows the change of friction coefficient with time under different loads. For A specimen, upon the test periods of 200 s, the friction coefficient increased with the increasing time, then fluctuated. When the load was 15 N and 20 N, the fluctuation trend was similar. The friction coefficient was stable after 900 s under the applied 25 N. However, for the B specimen, the friction coefficient was in a state of fluctuation before 900 s. It was stable under the load of 15 N. The friction coefficient increased slowly with time under the load of 20 N and 25 N, which indicated the transition from material intact to materials fracture [26].

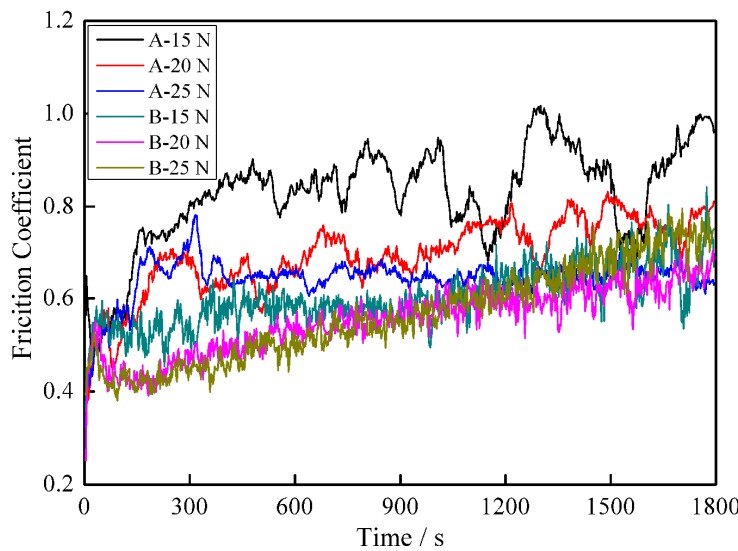

**Figure 6.** The change of friction coefficient of coatings A and B with time under different loads.

Figure 7 shows the average friction coefficients of the welding layers under different loads. It shows that the average friction coefficients presented a general decline with the increased loads. In general, as the load increased, the actual contact area increased, which was not the linear relationship with the increase of the coefficient. The average friction coefficient of the B specimen was lower than that of the A specimen. The wear mechanisms of CoCrFeMnNiW medium-entropy coatings would be interpreted based on the wear morphologies, and would be explained in detail below.

Figure 8 shows the wear rates of coatings A and B with increasing loads. It can be seen that the wear rate of coating A decreased from 0.244 mm$^3$·N$^{-1}$·m$^{-1}$ to 0.145 mm$^3$·N$^{-1}$·m$^{-1}$ and that of coating B decreased from 0.638 mm$^3$·N$^{-1}$·m$^{-1}$ to 0.605 mm$^3$·N$^{-1}$·m$^{-1}$ when the loads increased from 15 N to 25 N.

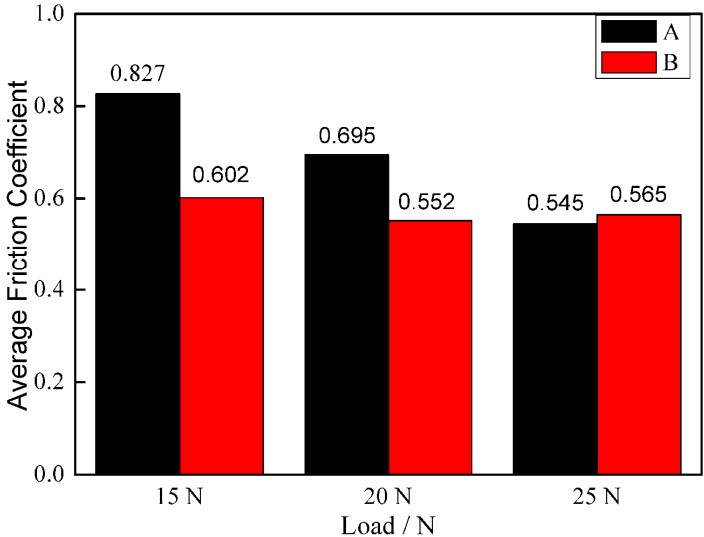

**Figure 7.** Average friction coefficients of the alloy coatings A and B under different loads.

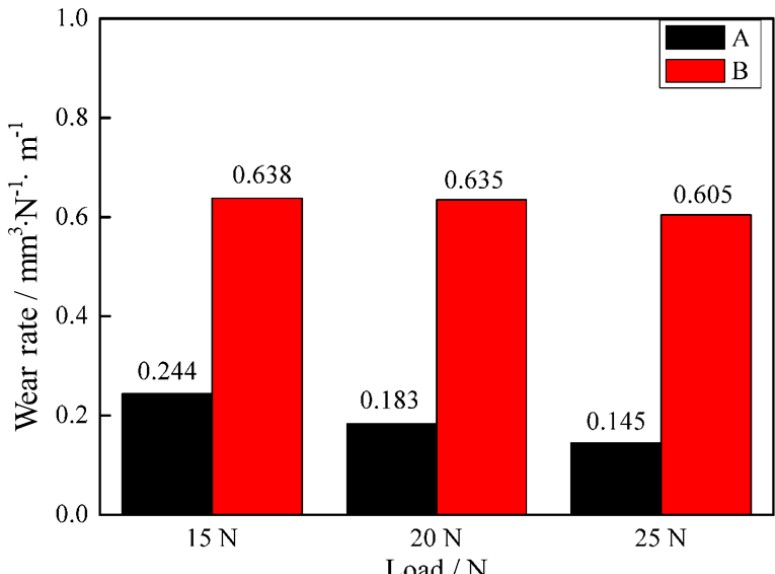

**Figure 8.** Wear rate of coatings A and B under different applied loads.

### 3.4. Wear Morphologies

In the initial wear stage, it was the contact between the surfacing layer and the opposite wear material. In the microscopic state, it was the contact between the friction pair and the convex surface body. Under the combined action of compressive stress and shear stress, the convex body felled off from the surface of the surfacing layer. As shown in Figure 9, obvious grooves existed on the worn surfaces of coatings A and B, which was the wear mechanism was abrasive wear. When the applied load increased, black adhesive appeared on the worn surface. Especially for coating B, the black adhesive became larger, and became sheet. The width of the wear scars increased as the applied load increased. Therefore, the wear mechanism of coating B was abrasive wear and adhesive wear.

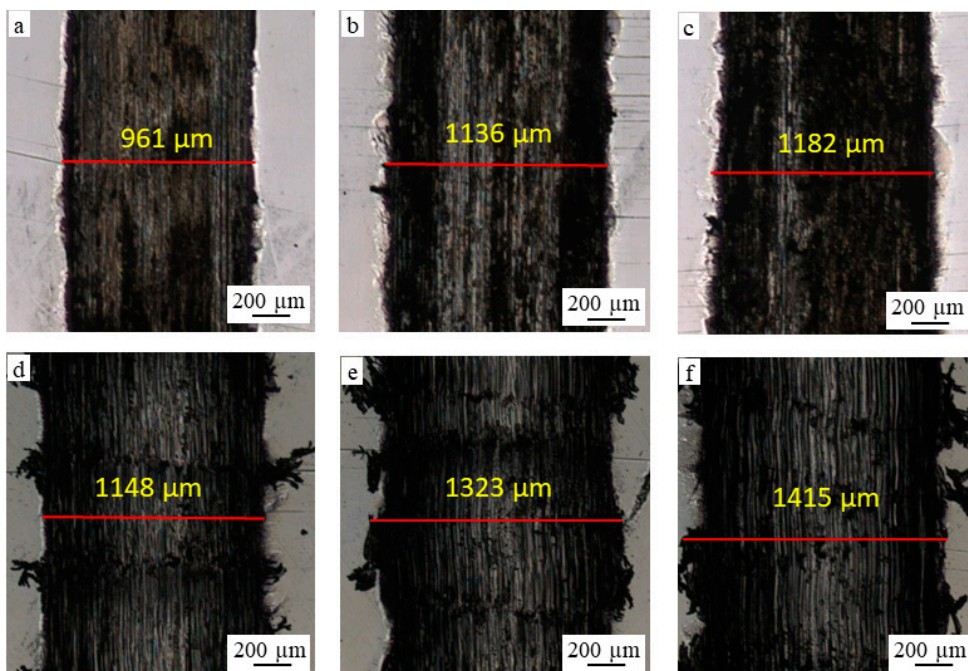

**Figure 9.** The wear scars under different loads with (**a**) 15 N, (**b**) 20 N and (**c**) 25 N; (**a–c**) is the images of coating A; (**d–f**) are the images of coating B.

Figure 10 shows the cross-sectional profiles of coatings A and B, under different loads. The depth of the coating A increased from 17 μm to 27 μm. It also can be seen that the depth and the width of the wear scar were essentially similar under loads of 20 N and 25 N. The depth of coating B increased from 46 μm to 68 μm, the width of the wear scar increased with the increasing loads. It was in very good agreement with the results shown in Figure 8.

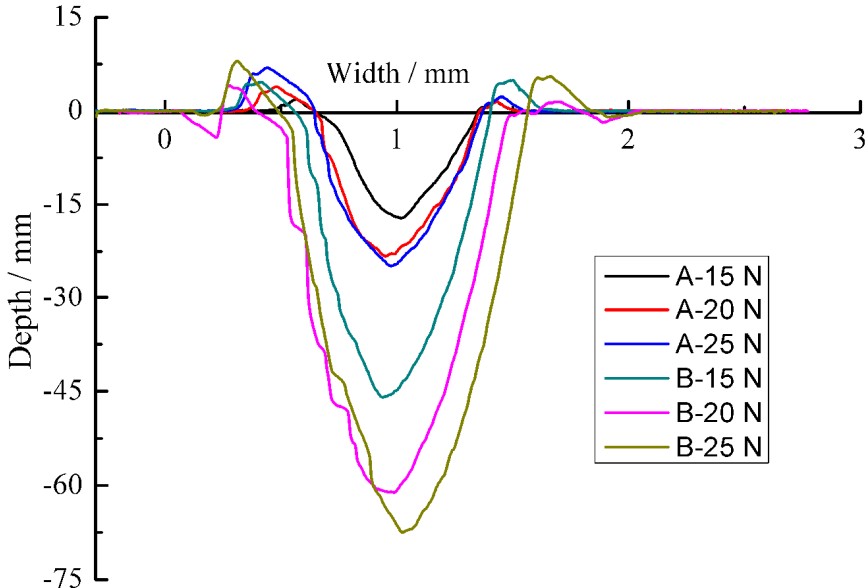

**Figure 10.** Two-dimensional (2D) profiles s of the cross-section of the wear scars.

Figure 11 shows the SEM micrographs of the worn surface under different loads. As can be seen from Figure 11a, there were a large number of grooves on the surface of coating A. There were many small white particles on the surface of the coating. They may be some hard particles or hard phases so that the corresponding friction coefficient and this

load were very high. This is consistent with the results in Figures 6 and 7. When the load was 20 N, delamination and tearing turned up in Figure 11b. There were mainly grooves that mean its wear mechanism was abrasive wear. This result was consistent with that of Figure 9. When the load got to 25 N, in Figure 11c, the main scratches were not only grooves but also delamination and tearing. It suggested that the wear mechanism was abrasive wear and fatigue wear. As shown in the energy spectrum results of the purple square area, there were some discontinuous oxidation films that had good anti-wear lubrication. It was the reason that its coefficient decreased in Figures 6 and 7. The obtained results show that A specimen had superior friction and wear performance. This is consistent with the analysis results in Figure 8.

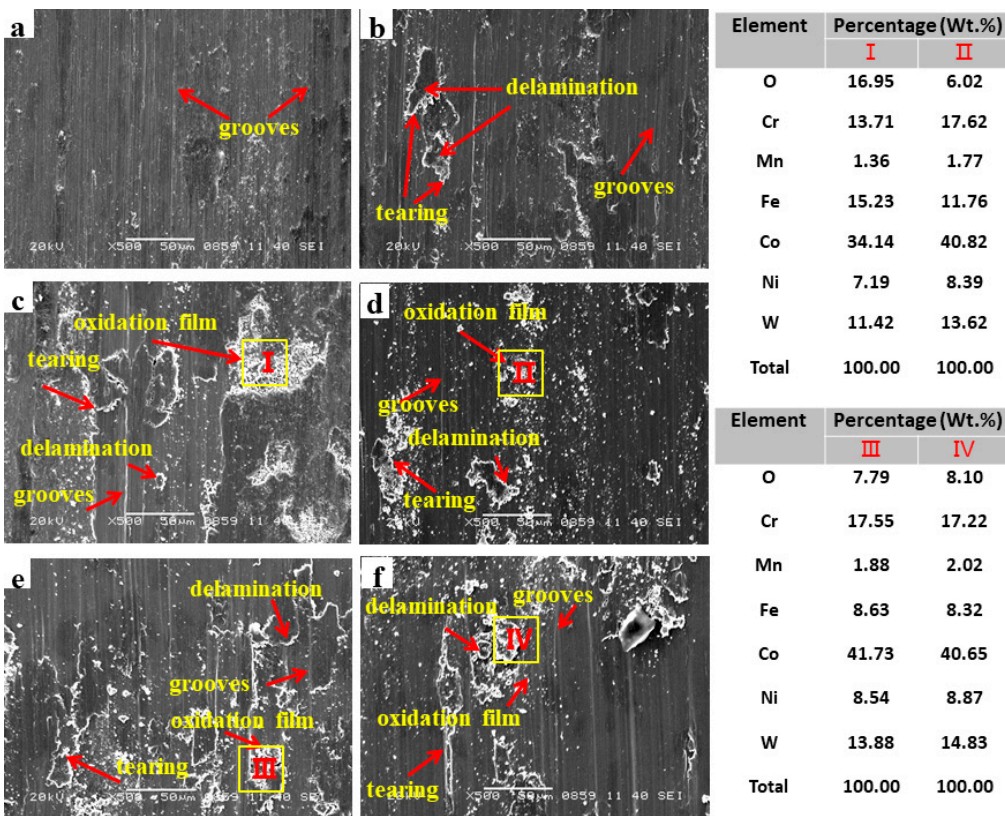

**Figure 11.** SEM micrographs of the worn surface under different loads a 15 N, b 20 N, c 25 N (**a–c**) is the images of coating A; (**d–f**) is the images of coating B.

　　From Figure 11d–f, there were many scratches with grooves, tearing and delamination on the surface of coating B under the load 15 N, 20 N and 25 N. When the load increased, the scratches became seriously, namely the grooves became deeper and the number of the delamination and tearing increased. During the surfacing welding process, thermal stress was produced that was easy to produced dislocations. The dislocations were the cause of the emergence of delamination. With the action of cyclic loading, dislocation accumulation under a certain depth of the surface produced micro-cracks, and they extended to the surface and connected to form delamination [27–31]. The existence of delamination indicated that fatigue wear was its wear mechanism. According to EDS analysis of the position, II, III and IV in Figure 11, it can be seen that the oxidation occurred under the load of 15 N, and oxidation also cured under the load of 20 N and 25 N. Discontinuous oxide film formed on the surface of the coating B. Because of the lubrication of the oxide film, the coating B had a smaller friction coefficient. Compared with the A coating, due to its smaller value of $\eta_h$ and EIT, the wear resistance of the coating B was weaker under the same load. According to the analysis of Figure 11d–f, the mechanism of the B specimen was abrasive in terms of wear and fatigue wear.

### 3.5. Vickers Hardness Measurements of the Coatings

Figure 12 shows the hardness profiles of the coating A and B. The hardness of coating A was lower than that of coating B. The test results were consistent with the analysis in Figure 4. The effect of alloying elements on the dilution ratio was studied by Cai and the result was that the hardness of coatings increased with increasing Fe content [20]. Obviously, this result is contrary to the findings published by Cai [20]. However, combined with Figure 6, the hardness of coating B increased and the change in wear rate was the opposite. This result is consistent with the results of Ye et al. [32].

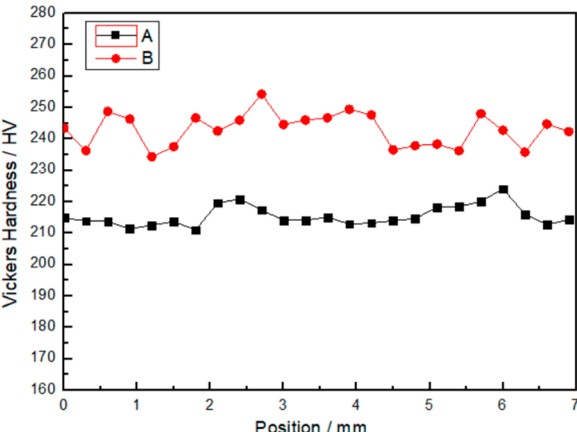

**Figure 12.** The surface hardness profiles of alloy coatings A and B.

### 4. Conclusions

The medium-entropy alloy CoCrFeMnNiW surfacing layer was prepared by plasma arc welding on Q235 surface. It was indicated that the MEA coating could be prepared by plasma arc technology by selecting appropriate filler wire and making full use of the dilution rate. However, the relationship between the optimal performance, dilution and process of the MEA coating was worth further discussion.

The microstructure of the alloy CoCrFeMnNiW surfacing layer was mainly divided into four parts from the bottom to the top of the surfacing layer. There was a fusion zone, equiaxed dendrites near the fusion zone, coarse columnar crystals and the near-surface with certain directions between the near-fusion zone and near-surface zone with fine equiaxed grains.

With the increase in the load, the alloy welding layer of the friction factor decreased. The wear mechanisms of the one-layer and two-layer coatings were different. The wear mechanism of the one-layer coating was abrasive wear and fatigue wear. The wear mechanism of the two-layer coating was adhesive wear and fatigue wear. For the CoCrFeMnNiW MEA coating, the main factors determining their wear resistance were the value of its depth recovery ratio ($\eta_h$) and EIT.

**Author Contributions:** Project administration, Q.H.; Data curation, X.W.; Formal analysis J.M.; Methodology and software, F.F. and X.S. All authors have read and agreed to the published version of the manuscript.

**Funding:** This research received no external funding.

**Institutional Review Board Statement:** Not applicable.

**Informed Consent Statement:** Not applicable.

**Data Availability Statement:** The data used to support the findings of this study are available from the corresponding author upon request.

**Conflicts of Interest:** The authors declare no conflict of interest.

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
