# Peer review of "Friction and Wear Performance of CoCrFeMnNiW Medium-Entropy Alloy Coatings by Plasma-Arc Surfacing Welding on Q235 Steel"

_coatings, doi:10.3390/coatings11060715_

Round 1

Reviewer 1 Report

The submitted manuscript concerns the wear resistance of CoCrFeMnNiW coatings. Please consider my comments:

  1. Abstract should contain a bit more information about the research results. The sentence "Moreover, the hardness was not the only factor of abrasion resistance." it's quite mysterious. Please write down what were the factors and not what was not the only factor.
  2. Try to avoid collective citation in the Introduction section
  3. The introduction is quite short. Perhaps it is worth introducing the reader to other methods of making such coatings. A brief introduction to the possibility of producing coatings e.g. using a laser would be advisable. There are quite a lot of papers described cobalt-based coatings produced using e.g. laser cladding.
  4. The authors cite recent work (2018-2021), which is a good side of the article.
  5. Line 48: "polished to mirror". In my opinion, this is not a scientific statement. Perhaps it is worth mentioning the roughness of the surface. It would be more appropriate in a research paper.
  6. Line 53: The caption for Figure 1 must be more detailed. Eg "Specimen areas dedicated for wear, microhardness and microstructure testing: (a) one-layer coating, (b) two-layer coating"
  7. Match the article to the journal's requirements. After specifying the name of the equipment, the country of origin must be written in brackets.
  8. The authors wrote on which apparatus the friction test was carried out. However, please write what movement was generated by the machine, what kind of friction is it?
  9. Line 85. In the caption of the table, please write that it is about the chemical composition. The information "composition" says nothing
  10. There are a lot of typos. Sometimes words are missing spaces. Please read the manuscript carefully.
  11. In my opinion, microhardness measurements should be carried out from the surface to the substrate. The X axis (position / mm) is not clear to me. Indicate the line on which the microhardness was measured on the microstructure.
  12. Another note on microhardness. The authors write about HV10 hardness. It is no longer microhardness. The microhardness ranges from HV0.01 to HV0.2. (according to some standards to HV0.5). Please describe how you measured hardness or microhardness, if it was tested.
  13. Please extend your conclusions.

The downside of the article, which is unacceptable in the current standards of writing papers, is the lack of references to the results of other researchers when discussing research results. If no such references are found, the novelty of the research should also be emphasized.

Author Response

Dear Editors//Reviewer,

We tried our best to improve the manuscript and made some changes in the manuscript.  These changes will not influence the content and framework of the paper. We appreciate for Editors/Reviewers’ warm work earnestly, and hope that the correction will meet with approval.
     Once again, thank you very much for your comments and suggestions.

Best regards.
                                             Xiaoli Wang

Reviewer 2 Report

Dear Authors,

I have read your paper "Friction and Wear Performance of CoCrFeMnNiW medium entropy alloys coating by Plasma arc Welding on Q235 steel" carefully.

This paper describes the process of the MEA fabricated and the result of the analysis properties of the coatings. 

The paper is easy to read.

But the methods are not properly described, so that other research groups may not reproduce them.

The paper is interesting. However, it requires few corrections.

  1. Please, improve the introduction. Why do you choose this alloy? 
  2. There is some mistake in the chemical composition of the Cobalt-based superalloy GH605. In table 1 and the Superalloy--Shenzhen Zhengjie Metal Materials Co., Ltd. (szzj168.com) are the different content of the Fe.
  3. Please, describe the equipment and modes of the welding process. In the text there is no value of the electrical power of the process fabrication.  
  4. Please, describe the purity of the weldin gases. 
  5. Please, capture "the substrate, coating" on the Fig 2 of the cross-section?
  6. Please, add EDS maps of the cross-sections one and two layers coatings (if possible).
  7. How many indentations were made? There is no information about place of the indentation. 
  8. Please specifically discuss the advantages of your work. Mark the main advantages compared to the other scientists (other methods). Now the discussion is poor.

The paper can be accepted for publication only after major improvements.

Author Response

(The authors gave the same response as above.)

Reviewer 3 Report

The review concerns the work entitled "Friction and Wear Performance of CoCrFeMnNiW medium entropy alloys coating by Plasma arc Welding on Q235 steel".

Main comments to the work are below:

1. The data in Table 1 (for Co) is incomprehensible. Please correct.
2. The section entitled "Materials and Methods" needs to be supplemented. Details of research methods and instruments are lacking. The Authors only mentioned devices such as UMT-2 or SEM, without specifying the tribologcal tests and measurement parameters.
3. Page 4, Figure 2. Please add the description of these images - what is what in the pictures.
4. Page 4, Figure 3. The legend does not fit this graph. Please enter the correct legend.
5. How many samples both A and B were tested? The minimum of three tests had to be carried out for each type of sample and each load. The data show that only one tribological test was taken for each load (Fig. 4 and Fig.8). So where does the average value in Figure 5 come from? If more tribological tests were performed for each material (A and B), then the standard deviation should be provided for each average test result.
6. Page 6, line 159. Incorrect citation of the drawing. It should be Figure 7 and not Figure 8.
7. Moreover, pleas check and correct the text, for example units - see page 3, line 75.

Author Response

(The authors gave the same response as above.)

Reviewer 4 Report

REVIEW OF  Friction and Wear Performance of CoCrFeMnNiW medium entropy alloys coating by Plasma arc Welding on Q235 steel

The paper has some serious flaws. The English has to be rewritten completely. Not only it has grammar mistakes, but also there are many incomprehensible sentences, which makes the manuscript impossible to understand.

The experiments are not well explain. The coating process was not well explained. The friction test was not explained. Most of the figures are of low quality.

In section 2 : The authors dont explain Q235

Equation 1 was not explained if applicable here.

In p3 : the 4 different parts are not explained, and not related to figures.

In p3 last paragraph: Why its extremely cold conditions? There’s no mention of cold condition elsewhere.

In P5: Why does the friction decrease with high load?

Fig 9: Most of the arrows do not explain and proof the notes mentions as : Groves, delamination tearing.

Overall many of the experimental controls are missing. And there are  no good explanation of the science behind the experiment and results.

Scientific Notes

  1. In page 2, line 70, what is this friction test? What is the main concept of it?
  2. In page 3, line 86, what is the entropy value? Give a small introduction for it?
  3. In page 3, line 92, “The entropy value of medium entropy alloys was from 92 0.693R to 1.609R” where is the reference and what does mean that the entropy value is medium?
  4. In page 4, line 121, the conclusion of the Nanoindentation test is “the A coating had a higher resistance to plastic deformation than the B coating”. Give a scientific explanation of this results. It can be related to material characterization.
  5. In page 5, line 137, It is not always the case that the friction coefficient is higher with an increased load. This result should be linked to this type of alloy.
  6. What is the material of the back adhesive layer mentioned in page 6, line 161? Is it from the alloy?
  7. Most of the results are based on observations. It will be better if it is based on observations linked to scientific explanations.

General Notes

  1. The abstract is not well written, and it has English mistakes. For Example:
  • “Moreover, the hardness was not the only factor of abrasion resistance”: this sentence is not understandable. The abstract should be very clear for the reader
  • The microstructure and performance of 10 the welding layer was researched by optical microscope”: this sentence has mistakes (were not was)
  1. The introduction is weak. It is not well written and not understandable. For example:
  • However, on the one the hand, there are few reports about MEA coating in plasma arc welding. On the other hand, MEA coating was prepared by melting powder method in many kinds of literature: this sentence is not understandable. What is the relation between the two.
  • The structure of the introduction is not sufficient. “Plasma arc welding is widely used in part surface abrasion resistant and corrosion 29 resistant alloy welding layer deposition of petroleum, mining and metallurgy, and other 30 areas of the condition because of high energy concentration and penetration” this sentence should be used at the beginning of the introduction
  1. Figure 3 quality is too bad. Is it a screenshot?

The paper is way below an acceptable level for publication.

Author Response

(The authors gave the same response as above.)

Round 2

Reviewer 1 Report

Please publish in its present form. 

Author Response

Dear editor,

We appreciate for Editors/Reviewers’ warm work earnestly, 

Once again, thank you very much for your comments and suggestions.
   Sincerely yours,

Authors

Reviewer 2 Report

Dear Authors,

I have read your modified paper "Friction and Wear Performance of CoCrFeMnNiW medium entropy alloys coating by Plasma arc Welding on Q235 steel" carefully.

The materials and methods are properly described, so that other research groups may reproduce them. Explanations are clear and the paper is easy to read.

I can recommend the Editor to accept this revised manuscript to be published in Coatings.

Author Response

Dear editor,

We appreciate for Editors/Reviewers’ warm work earnestly.

Once again, thank you very much for your comments and suggestions.
   Sincerely yours,

Authors

Reviewer 3 Report

Please check the quality (readable) of Figure 7 and correct the blue fonts.

Reviewer 4 Report

Revision Two:

Friction and Wear Performance of CoCrFeMnNiW medium entropy alloys coating by Plasma arc Welding on Q235 steel

Scientific Notes:

Even though many changes were made, the paper still have some serious flaws:

  1. In page 3, line 96. It was mentioned that the experiments were repeated five times, and the average was taken. Was there a big difference between one time and another?
  2. In page 7 friction coefficient are shown, but there is no mention about the experimental details.
  3. In page 5, line 136, what does the entropy difference tell, what does it mean one coating has a higher entropy than the other? Clarify this point.
  4. Clarify the of high load in the friction test gives high friction factor? Put a reference for it.
  5. Fatigue wear is mentioned many places, but there is no proof or evidence about it.
  6. What is EMA in Page 2?
  7. Why are the authors mentioning welding passes in section 2, was is plasma arc or welding?
  8. In Fig 3: there is no clear evidence of dendrites.
  9. In page 5: why does W selectively change the matrix lattice?
  10. Why is the hardness of two layer higher, where are the measurements?
  11. In Fig 4, why are there so many differences in alloying elements, maybe the process is not well controlled.
  12. in page 5 where was the friction couple experiments ?? why there is no explanation about it?
  13. In page, 7 the wear reduction has to be measured and indicated before.
  14. Fig 11, it is not easy to read the tables.
  15. The wear mechanism and types are not explained and are not justified by Fig 11.

General Notes:

  1. The English has so many mistakes (for example in page 2 line 62 the sentence is not complete: When the first welding pass 1 was finished and cooled in the atmosphere for 5 min, and then the second welding pass 2 was started in the same direction).
  2. Some experiments are still not well explained. For example, the coating process was not well explained. The friction test also was not explained.
  3. Most of the figures are of low quality.

Overall, the quality of the paper is not on the level of a journal. The experiments are not clear, the findings are not justified, and the science is lacking when relating the finding to physical science or phenomena.
